# KineDiff3D: Kinematic-Aware Diffusion for Category-Level Articulated Object Shape Reconstruction and Generation

## Abstract

Articulated objects, such as laptops and drawers, exhibit significant challenges for 3D reconstruction and pose estimation due to their multi-part geometries and variable joint configurations, which introduce structural diversity across different states. To address these challenges, we propose **KineDiff3D**: Kinematic-Aware Diffusion for Category-Level Articulated Object Shape Reconstruction and Generation, a unified framework for reconstructing diverse articulated instances and pose estimation from single view input. Specifically, we first encode complete geometry (SDFs), joint angles, and part segmentation into a structured latent space via a novel Kinematic-Aware VAE (**KA-VAE**). In addition, we employ two conditional diffusion models: one for regressing global pose (SE(3)) and joint parameters, and another for generating the kinematic-aware latent code from partial observations. Finally, we produce an iterative optimization module that bidirectionally refines reconstruction accuracy and kinematic parameters via Chamfer-distance minimization while preserving articulation constraints. Experimental results on synthetic, semi-synthetic, and real-world datasets demonstrate the effectiveness of our approach in accurately reconstructing articulated objects and estimating their kinematic properties.

## 1 Introduction

Articulated objects, like laptops, drawers, and robotic arms, are common in everyday and industrial settings, featuring multi-part geometries linked by joints that enable diverse motions. Reconstructing their 3D geometry and kinematic structure from a single view input is essential for robotics Eisner et al. (2022), augmented reality Jiang et al. (2022a), and computer vision Song et al. (2024). However, this task is challenging due to the complex structures and joint-induced degrees of freedom, which create numerous possible

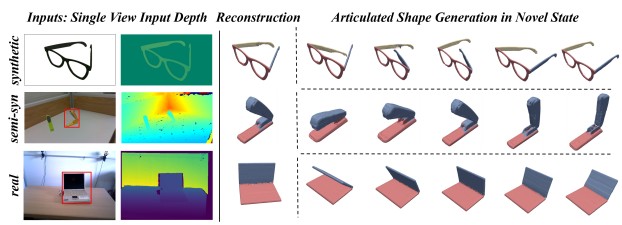

Figure 1: **KineDiff3D demonstrates strong generalization across datasets** (synthetic, semi-synthetic, real-world), achieving robust **reconstruction** and **novel articulated shape generation** from single-view depth inputs.

configurations. This complexity makes it difficult to infer complete 3D structures and kinematic parameters from partial or occluded views. Moreover, accurately estimating the pose of the object and the states of its joints is crucial for understanding its functionality and interaction capabilities, adding further complexity to the task.

Existing methods Mandi et al. (2024); Liu et al. (2023); Jiang et al. (2022b) struggle to balance efficiency and generalization when addressing these issues. Instance-specific approaches based on multi-view observations can reconstruct detailed object models but typically require complete multi-view coverage or known articulation priors to achieve good results. These methods exhibit poor generalization to unseen instances within the same category (e.g., eyeglasses of different models)

and are highly sensitive to occlusion; even minor occlusion or missing viewpoints often prevents the recovery of complete geometry. Category-level methods, which learn shape priors for object categories (e.g., A-SDF's Mu et al. (2021) encoder for single-view inference), face critical challenges: directly regressing the signed distance fields (SDFs) results in overly smooth reconstructions lacking fine details; their insufficient modeling of articulation fails to ensure part coherence under complex joint interactions; and they heavily depend on precise canonical pose alignment before reconstruction. More critically, these methods typically treat shape reconstruction and pose estimation as two independent stages, neglecting the tight coupling between them, which can consequently lead to suboptimal final results. In real-world scenarios, an object's geometry and its pose are complementary; considering them simultaneously is essential to achieve more accurate reconstruction outcomes.

To address the aforementioned challenges, we propose **KineDiff3D**: Kinematic-Aware Diffusion for Category-Level Articulated Object Shape Reconstruction and Generation. KineDiff3D provides a unified framework capable of simultaneously recovering an object's complete geometry, kinematic configuration, and generating novel articulated states (i.e., new joint configurations for the same object instance) from a single-view input (see Figure 1). Our approach comprises the following three core, synergistic modules: 1)Kinematic-Aware Shape Prior Learning: We train a Kinematic-Aware Variational Autoencoder (KA-VAE) to encode the complete object's SDF values, joint angles, and part segmentation masks into a compact latent code Z. This integrated representation captures essential geometric and kinematic properties, establishing a prior where latent interpolations yield smooth transitions in both geometry and joint states. Crucially, this prior enables the generation of diverse, kinematically valid configurations by manipulating the joint angles within the latent space. 2) Diffusion-Based Pose Estimation and Shape Reconstruction Modules: To tackle the challenges of pose ambiguity and partial observation, we design two synergistic conditional diffusion models operating within a joint-centric framework during both training and inference. The first diffusion model conditions on the input partial point cloud and is dedicated to robustly estimating the object's global pose and joint parameters. The second diffusion model leverages the estimated pose to transform the partial observation into a canonical space and conditions on this normalized input. It then progressively generates the kinematic-aware latent code Z from noise. This collaborative mechanism bridges partial observations to a complete model representation, enabling full structural recovery. 3) Iterative Joint Centric Optimization: During inference, we incorporate an iterative optimization module based on Chamfer distance to jointly refine the estimated global pose, joint parameters and reconstructed geometric shape. This joint-centric process bidirectionally minimizes the distance between the transformed reconstructed mesh and the input partial point cloud while strictly preserving the articulation constraints and part connectivity defined by the kinematic structure. This iterative loop significantly enhances both reconstruction accuracy and detail recovery.

Our main contributions can be summarized as follows: (1) a novel unified framework based on diffusion models that jointly addresses category-level reconstruction, pose estimation, and generation of articulated objects from single-view inputs; (2) a Kinematic-Aware Shape Prior (KA-VAE) that co-encodes geometry (SDFs), joint angles, and part segmentation into a structured latent space, enabling continuous interpolation of shape and articulation and serving as the foundation for novel shape generation; (3) a Joint-Centric Iterative Optimization strategy that bidirectionally refines global pose, joint parameters, and geometry via Chamfer distance minimization, strictly preserving articulation constraints to boost accuracy and physical plausibility; (4) extensive experiments show that our method achieves favorable performance in category-level reconstruction and pose estimation of articulated objects compared to existing methods.

## 2 RELATED WORK

### 2.1 3D RECONSTRUCTION OF ARTICULATED OBJECTS

Recent approaches like Paris Liu et al. (2023), Real2Code Mandi et al. (2024), Ditto Jiang et al. (2022b) have demonstrated significant advancements in reconstructing articulated objects, they often depend on dense supervision or multiple views, which may not be available in real-world scenarios. Similarly, emerging techniques such as DigitalTwinArt Weng et al. (2024), ArtGS Liu et al. (2025), and ArticulatedGS Guo et al. (2025), all rely on multi-view inputs and advanced rendering techniques. While these methods achieve high-fidelity reconstructions, their practicality is limited in scenarios where only a single view is available. Methods like A-SDF Mu et al. (2021) and CARTO

Heppert et al. (2023) have proven the feasibility of learning from single-view input by training an encoder to extract shape and articulation features into a latent space, which is then decoded to predict SDF values at query points. However, a critical limitation of these approaches lies in their direct regression of SDF values, this strategy often struggles to recover high-fidelity geometric details and sharp features, resulting in overly smoothed reconstructions with reduced precision, particularly near articulation joints and part boundaries.

In contrast, KineDiff3D operates directly on partial point clouds derived from single-view inputs. It employs a KA-VAE to jointly encode complete object SDFs, joint angles, and part segmentation into a unified latent space. This integrated representation enables category-level generalization without requiring dense supervision, while significantly enhancing both the modeling efficiency and reconstruction quality for articulated objects.

## 2.2 Diffusion Models for 3D Reconstruction

Diffusion models have emerged as powerful tools for generative tasks in 3D computer vision, especially for reconstructing objects from partial or noisy inputs. Zhou & Tulsiani (2023); Wang et al. (2025); Chen et al. (2024) uses a diffusion process to generate 3D point clouds from partial observations, achieving high-quality reconstructions for rigid objects. Similarly, Chou et al. (2023); Shim et al. (2023) applies diffusion models to generate SDFs for static objects, conditioned on partial point clouds or images. These approaches demonstrate the ability of diffusion models to capture complex distributions of 3D shapes, but they are primarily designed for rigid objects and do not account for articulated objects.

For articulated objects, diffusion models are less explored. Recent work by Cheng et al. (2024) adapts diffusion models to generate articulated hand poses, conditioned on partial 2D observations. However, this approach focuses on human hands and does not generalize to articulated objects with diverse kinematic structures. KineDiff3D extends the application of diffusion models to articulated objects by conditioning the model on partial 3D observations. Our framework generates a latent code that encapsulates the object's SDFs, joint angles, and part segmentation while estimating global pose and joint parameters, addressing the unique challenges of articulated object reconstruction.

## 3 Methodology

### 3.1 Kinematic-Aware Shape Prior Learning

In the field of 3D reconstruction, articulated objects present unique challenges due to complex multi-part geometries and variable joint configurations, traditional methods Park et al. (2019); Wang et al. (2021); Melas-Kyriazi et al. (2023) struggle to handle the significant geometric variations exhibited by the same instance under different joint states (e.g., open/closed laptops). To address this, we propose a Kinematic-Aware Variational Autoencoder (KA-VAE) that jointly encodes geometric representations (SDFs), joint angles $A \in \mathbb{R}^{K-1}$ (where K is the number of rigid parts), and part segmentation labels $S \in \{0, 1, \ldots, K-1\}^N$ into a unified latent space, simultaneously modeling surface geometry and dynamic joint angles. This design achieves two core objectives: (1) establishing a continuous latent space where latent vector interpolations correspond to smooth geometric transitions and continuous joint angle variations; (2) ensuring diversity to capture geometric features of various instances within the same category, ultimately constructing a robust representation that enables diffusion models to efficiently learn and bidirectionally map geometric-motion properties.

To this end, our architecture consists of a PointNet encoder $\Phi$, a Kinematic-Aware Variational Autoencoder (KA-VAE) $\Theta$, a multilayer perceptron (MLP) $\Psi$, and an Articulation decoder $\Omega$ (see Figure 2). The KA-VAE encodes geometric information into a unified latent vector $Z$, while the Articulation decoder $\Omega$ conditions on $Z$ and joint angles $A$ to produce flexible generations. Specifically, the input point cloud $P \in \mathbb{R}^{N \times 3}$ is processed by the PointNet encoder $\Phi$ to extract high-dimensional geometric features $F_g \in \mathbb{R}^d$. These features are fed into the KA-VAE encoder $\Theta_{\text{enc}}$, which generates the mean $\mu \in \mathbb{R}^{D_z}$ and variance $\sigma^2 \in \mathbb{R}^{D_z}$ of a latent distribution, from which the latent vector $Z \sim \mathcal{N}(\mu, \sigma^2)$ is sampled using the reparameterization trick Doersch (2021). To enable kinematic-geometric fusion, joint angles $A \in \mathbb{R}^{K-1}$ are encoded by the MLP $\Psi$ into a kinematic feature vector $\alpha \in \mathbb{R}^{D_z}$. The vector $\alpha$ is then concatenated with $Z$, and the combined input $concat(Z, \alpha)$ is passed to the KA-VAE decoder $\Theta_{\text{dec}}$ to reconstruct a kinematic-aware geometric

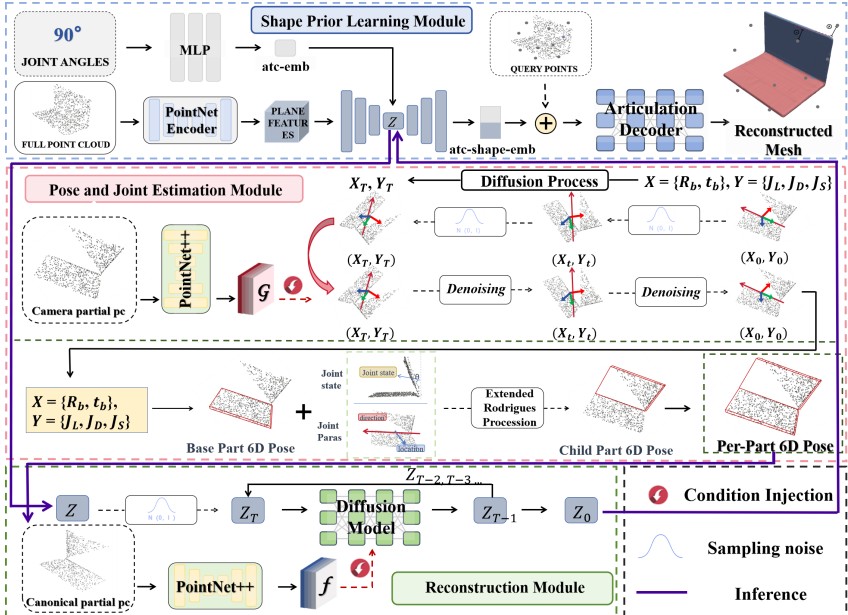

Figure 2: **Pipeline of the KineDiff3D Framework.** This framework integrates three synergistic modules: **(a) Shape Prior Learning Module (top)**: Given a full object point cloud and its joint angles, a PointNet-based KA-VAE learns a shape prior embedding. **(b) Pose and Joint Estimation Module (center)**: Employs a conditional diffusion model to jointly optimize base pose (X) and joint parameters (Y) during the denoising process. **(c) Reconstruction Module (bottom)**: Learns the KA-VAE encoded shape prior embedding through diffusion modeling, then combines angle encoding with KA-VAE decoding via a kinematics-constrained articulation decoder to generate reconstructed articulated meshes.

feature $F_{kg} = \Theta_{\text{dec}}(concat(Z, \alpha))$. The Articulation Decoder $\Omega$, comprising an SDF network and task-specific heads, takes $F_{kg}$ as input and predicts SDF values $\hat{\text{SDF}}_Q \in \mathbb{R}^L$, part segmentation labels $\hat{S} \in \{0, 1, \ldots, K-1\}^N$, and joint angles $\hat{A} \in \mathbb{R}^{K-1}$. This enables the model to reconstruct the object's geometry and kinematic state for any specified joint configuration. The encoding and decoding process is mathematically described as:

$$\begin{cases} F_g = \Phi(P), \quad \left[\mu, \sigma^2\right] = \Theta_{\text{enc}}\left(F_g\right), \\ Z \sim \mathcal{N}\left(\mu, \sigma^2\right), \quad \alpha = \Psi(A), \\ F_{kg} = \Theta_{\text{dec}}\left(\text{concat}(Z, \alpha)\right), \\ \left[\hat{\text{SDF}}_Q, \hat{S}, \hat{A}\right] = \Omega\left(Q | F_{kg}\right) \end{cases} \tag{1}$$

where $Q \in \mathbb{R}^{L \times 3}$ is the concatenation of query points.

To ensure the latent vector $Z$ satisfies the requirements of diffusion models, we regularize the latent space using KL divergence and employ multi-task learning to predict SDF values, segmentation labels, and joint angles. The training objective is defined as:

$$\mathcal{L}_{\text{KA}} = \lambda_1 \|\text{SDF}_Q - \hat{\text{SDF}}_Q\|_1 + \lambda_2 \mathcal{L}_{\text{CE}}(S, \hat{S}) + \lambda_3 \|A - \hat{A}\|_1 + \beta D_{\text{KL}}\left(\mathcal{N}(\mu, \sigma^2) \| \mathcal{N}(0, 0.25^2)\right) \tag{2}$$

The fourth term in Equation 2 imposes KL divergence regularization, constraining the latent distribution $\mathcal{N}(\mu, \sigma^2)$ to approximate a Gaussian prior $\mathcal{N}(0, 0.25^2)$. This regularization serves two critical purposes: 1) enforcing a structured latent space where proximity implies similarity in geometric and kinematic configurations, ensuring smooth interpolations; and 2) aligning the latent distribution with the Gaussian noise injection in diffusion model training. The prior's standard deviation of 0.25 balances diversity capture (across object instances) and compatibility with the diffusion noise schedule. The hyperparameter $\beta = 10^{-3}$ controls regularization strength, while $\lambda_1 = 1, \lambda_2 = 0.1, \lambda_3 = 0.1$ balance loss components. Collectively, these terms enable the KA-VAE to learn robust, expressive latent representations suitable for downstream diffusion-based reconstruction.

## 3.2 POSE AND JOINT ESTIMATION MODULE

Estimating the pose of articulated objects from a single view is challenging due to complex structures and joint-induced degrees of freedom. Traditional pose estimation methods Li et al. (2020); Liu et al. (2022b) for articulated objects often adopt a *part-centric* approach, independently estimating the pose of each part. Such methods overlook inherent kinematic constraints and struggle with severe self-occlusions. To address these limitations, we adopt a *joint-centric* strategy. We consider an articulated object with K rigid parts, this approach leverages the kinematic tree structure: we estimate the global SE(3) pose $\{R_{base}, t_{base}\}$ of the base part (which moves freely) and the parameters defining the joints connecting parent and child parts. Specifically, joints are modeled as either: (1) Revolute Joints, parameterized by a joint state $\theta_r \in \mathbb{R}$, joint location $\mathbf{l}_r \in \mathbb{R}^3$, and joint direction $\mathbf{d}_r \in \mathbb{R}^3$; or (2) Prismatic Joints, parameterized by a displacement $\theta_p \in \mathbb{R}$ and direction $\mathbf{d}_p \in \mathbb{R}^3$. The poses $\{\{R_{\text{child}}^{(k)}\}_{k=2}^K, \{t_{\text{child}}^{(k)}\}_{k=2}^K\}$ for K-1 child parts are then deterministically derived using kinematic chaining. Crucially, to capture the multi-modal nature of possible configurations consistent with the partial observation $O$, we formulate pose estimation as sampling from the conditional data distribution $p_{data}(x|O)$, where $x$ represents the state parameters. We use a conditional diffusion model for this task, as it excels in modeling complex, multi-modal distributions and generating diverse, plausible hypotheses from observations.

We represent the state $x$ to be estimated as a vector encompassing the base part's pose. We represent rotation $R_{base}$ as a continuous 6D vector $\mathbf{r}_{6D} = [\mathbf{a}_1^\top, \mathbf{a}_2^\top]^\top \in \mathbb{R}^6$ (the first two columns of $R_{base}$) avoiding SO(3) discontinuities. This is concatenated with translation $t_{base} \in \mathbb{R}^3$ to form the state parameters $x = [\mathbf{r}_{6D}^\top, \mathbf{t}_{\text{base}}^\top]^\top \in \mathbb{R}^9$.

Specifically, we first adopt the Variance-Exploding (VE) Stochastic Differential Equation (SDE) Song et al. (2021) to construct a continuous diffusion process $\{x(t)\}_{t=0}^1$, indexed by the time variable $t \in [0, 1]$. As t increases from 0 to 1, the time-indexed pose variable $x(t)$ is perturbed by the following SDE function:

$$dx = g(t)dw, \quad \text{where} \quad g(t) = \sqrt{\frac{d[\sigma^2(t)]}{dt}} \tag{3}$$

where $\sigma(t)$ a time-varying hyper-parameter, $g(t)$ is the diffusion coefficient, and $w$ is a standard Wiener process.

During training, a score model $\mathbf{s}_\Theta(x(t), t|O)$ Song & Ermon (2019) is trained to approximate the score function $\nabla_x \log p_t(x|O)$. The score model is optimized using Denoising Score Matching (DSM). Given samples $x(0) \sim p_{data}(x(0)|O)$, corresponding noisy samples $x(t) \sim \mathcal{N}\left(x(t); x(0), \sigma^2(t)I\right)$ are generated by solving the forward SDE. The training objective minimizes:

$$\mathcal{L}(\Theta) = \mathbb{E}_{t \sim \mathcal{U}(\delta,1)} \left\{ \lambda(t) \mathbb{E} \left[ \left\| \mathbf{s}_\Theta(x(t), t|O) - \frac{x(0) - x(t)}{\sigma(t)^2} \right\|_2^2 \right] \right\} \tag{4}$$

where $\delta$ is a hyper-parameter representing the minimum noise level. At inference, we can approximately sample pose $\hat{x}$ from $p_{data}(x|O)$ by sampling from $p_\delta(x|O)$, as $\lim_{\delta \to 0} p_\delta(x|O) = p_{\text{data}}(x|O)$. To sample from $p_\delta(x|O)$, we can solve the following Probability Flow (PF) ODE Song et al. (2020) where $x(1) \sim \mathcal{N}(0, \sigma^2 I)$, from $t = 1$ to $t = \delta$: $dx = -\sigma(t)\dot{\sigma}(t)\nabla_x \log p_t(x|O)dt$. The score function $\log p_t(x|O)$ is empirically approximated by the estimated score network $\mathbf{s}_\Theta(x(t), t|O)$ and the ODE trajectory is solved by RK45 ODE solver Dormand & Prince (1980).

Note that the joint parameters are predicted using the same way as the base part pose. Child-part poses are then computed from the estimated base-part pose and joint parameters using kinematic chaining and the Rodrigues formula: $R_{child} = R_{base}(I \cdot \cos\theta + (1 - \cos\theta) \cdot (d \cdot d^T) + W \cdot \sin\theta)$, Where $I$ is the identity matrix, $\theta$ is the joint state. The matrix $W$ is the skew-symmetric matrix of $d$. The translation vector $t_{child}$ is computed as the distance between the center coordinate of point cloud $c$ and the rotated coordinates $R_{hild} \cdot c$, formulated as $t_{child} = c - R_{child} \cdot c$.

## 3.3 SHAPE RECONSTRUCTION AND GENERATION MODULE

Building upon the KA-VAE's structured latent space, we now address the core reconstruction challenge: predicting the kinematic-aware latent vector $Z$ from incomplete single-view observations.

For articulated objects like laptops or drawers, we typically only observe partial point clouds in arbitrary configurations. To tackle this, we design a conditional diffusion model that progressively refines noise into meaningful latent representations, directly guided by the partial observation throughout the denosing trajectory. This diffusion process acts as a probabilistic "inverse solver" that maps sparse inputs to the complete geometric-joint encoding captured by $Z$. During training, we transform the observed partial point cloud from camera space to canonical space using ground-truth (GT) poses, resulting in normalized inputs $C \in \mathbb{R}^{N \times 3}$ that serve as the conditioning signal.

The diffusion framwork operates through two interwined phases: *a forward noising process* that systematically corrupts clean latent vectors, *and a conditional reverse process* that reconstructs them using partial observations as guiding signals. Instead of predicting the added noise $\xi$ as in the original DDPM Ho et al. (2020), we follow Ramesh et al. (2022) and predict $Z_0$, the original, denoised vector. Through T iterative steps, we gradually add Gaussian noise according to predefined variance schedule $\{\beta_t\}_{t=1}^{T}$, transforming $Z_0$ into increasingly noisy versions $Z_1, Z_2, \ldots, Z_T$ until it becomes pure noise. Mathematically, this forward diffusion follows:

$$q(Z_t|Z_{t-1}) = \mathcal{N}\left(Z_t; \sqrt{1-\beta_t}Z_{t-1}, \beta_t I\right) \tag{5}$$

This process can be expressed in closed form as:

$$Z_t = \sqrt{\bar{\alpha}_t}Z_0 + \sqrt{1-\bar{\alpha}_t}\boldsymbol{\xi}, \quad \boldsymbol{\xi} \sim \mathcal{N}(0, I) \tag{6}$$

with $\bar{\alpha}_t = \prod_{s=1}^{t}(1-\beta_s)$. By step $T$, $Z_T$ is approximately pure noise, $Z_T \sim \mathcal{N}(0, I)$.

The reverse diffusion process learns to denoise $Z_t$ back to $Z_0$, conditioned on the partial point cloud $C \in \mathbb{R}^{N \times 3}$. At each timestep $t$, a denosing network $\epsilon_\theta$ directly predicts the denoised latent code $Z_0$, while cross-referencing geometric features extracted from $C$. Specifically, a lightweight PointNet++ Qi et al. (2017) enoder $\Gamma$ processes $C$ into a condition feature vector $F_{cond} = \Gamma(C)$. This vector guides $\epsilon_\theta$ through cross-attention layers Vaswani et al. (2017), where the noisy latent $Z_t$ serves as the query, and $F_{cond}$ provides the keys and values. The cross-attention mechanism computes attention weights to align $Z_t$ with the geometric information in $F_{cond}$, ensuring that each denoising step refines $Z_t$ in a manner consistent with the observed partial point cloud. This dynamic alignment enables the model to generate structurally coherent completions. The network is trained to minimize the $Z_0$ reconstruction error:

$$\mathcal{L}_{\text{diff}} = \|\epsilon_\theta\left(Z_t, t, \Gamma(C)\right) - Z_0\|_2 \tag{7}$$

During test time, reconstruction begins by sampling pure noise $Z_T \sim \mathcal{N}(0, I)$. We then iteratively denoise it over $T$ steps, with each step $t$ refining $Z_t$ into $Z_{t-1}$ using the denosing network $\epsilon_\theta$:

$$Z_{t-1} = \epsilon_\theta\left(Z_t, t, \Gamma(C)\right) + \sigma_t\xi, \quad t = T, T-1, \ldots, 1 \tag{8}$$

where , $\xi \sim \mathcal{N}(0, \mathbf{I})$ and $\sigma_t$ is the fixed standard deviation at the given timestep. We iteratively denoise $Z_T$ until we obtain the final output $Z'$. Then, we pass the generated latent vectors $Z'$ back into the Articulation-VAE model to reconstruct a complete, articulation-aware 3D model.

Crucially, this latent representation enables dynamic shape generation: by retaining $Z'$ while modifying joint angles $A$ through the kinematic feature vector $a = \Psi(A)$, we synthesize novel configurations via $[\hat{\text{SDF}}_Q, \hat{S}, \hat{A}] = \Omega(Q|(concat(Z', a))$. This joint-aware generation allows real-time articulation manipulation Yu et al. (2024), like rotating laptop screens or translating drawers without geometry recomputation. The reconstruction-to-synthesis workflow executes in one pass through the disentangled latent space.

### 3.4 INFERENCE AND ITERATIVE OPTIMIZATION

The KineDiff3D framework processes single-view partial point clouds in camera space to jointly estimate pose, joint parameters, and reconstruct complete articulated geometry. The inference pipeline operates as follows:

**Initial Pose and Joint Estimation.** Given an input partial point cloud $O \in \mathbb{R}^{N \times 3}$ in camera space, the trained score model predicts the base part pose and joint parameters. Child part poses are derived via kinematic chaining using the Rodrigues formula, ensuring adherence to articulated constraints.

**Canonical Transformation and Latent Diffusion.** The input $O$ is transformed into canonical space using the estimated poses, yielding a normalized partial point cloud $C$. This canonicalized observation conditions the Latent diffusion model, which generates the kinematic-aware latent vector Z. The Articulation decoder $\Omega$ then reconstructs the complete SDF values, segmentation masks, and joint angles from Z, enabling extraction of a watertight mesh $\mathcal{M}$.

**Iterative Optimization via Joint Centric Chamfer Alignment.** Initial estimates of pose and geometry may be suboptimal due to occlusion or ambiguity. To refine results, we introduce a joint-centric optimization loop minimizing the bidirectional Chamfer distance $\mathcal{L}_{CD}$ between the transformed reconstructed mesh $\mathcal{M}$ and input $O$ (see Figure 3):

Figure 3: **Kinematic Iterative Optimization Pipeline.** Joint-Centric Chamfer Alignment iteratively refines SE(3) part poses, joint parameters and reconstruction precision.

$$\mathcal{L}_{\text{CD}} = \sum_{k=1}^{K} \text{Chamfer}\left(\mathcal{M}^{(k)} \cdot T^{(k)}, O^{(k)}\right) \qquad (9)$$

where $T^{(k)} = \{R^{(k)}, t^{(k)}\}$ denotes the pose of the k-th part, and $O^{(k)}$ is the corresponding partial point cloud. Crucially, child poses $T_{\text{child}}^{(k)}$ remain functionally dependent on base pose $T_{\text{base}}$ and joint parameters through kinematic constraints (e.g., $T_{\text{child}} = f(T_{\text{base}}, l_r, d_r, \theta_r)$). This dependency ensures part connectivity is preserved during gradient-based updates.

The optimization alternates between: **1.Pose Refinement:** Adjusting $T_{base}$ and joint parameters to minimize $\mathcal{L}_{CD}$ via gradient descent. **2.Geometry Reconstruction:** Updating the mesh $\mathcal{M}$ by regenerating $Z$ conditioned on the latest canonicalized point cloud $C'$.

## 4 EXPERIMENTS

### 4.1 DATASETS

We conducted our experiments using two datasets: the synthetic dataset ArtImage Xue et al. (2021) and the semi-synthetic dataset ReArtMix Liu et al. (2022a). Following the Mu et al. (2021), we generated SDF samples for each articulated shape across both datasets. Simultaneously, we annotated part segmentation masks, 6D poses, and joint parameters to provide comprehensive ground truth for reconstruction and pose estimation tasks. See supplementary material for detailed statistics.

### 4.2 BASELINES AND METRICS

**Reconstruction and Generation Task.** We benchmark our approach against state-of-the-art category-level reconstruction methods most relevant to our work: A-SDF Mu et al. (2021) and CARTO Heppert et al. (2023). Additionally, to further evaluate against methods requiring multiview inputs, we include Paris Liu et al. (2023) and Ditto Jiang et al. (2022b) with 16 views for articulated object reconstruction. Since the original implementations of A-SDF, Paris, and Ditto lack pose estimation capabilities, we augment all three with full ground-truth pose data. For evaluation metrics, we employ the Chamfer-L1 distance (CD) to measure reconstructed mesh quality: CD-w quantifies overall surface reconstruction accuracy, while CD-s and CD-m separately measure reconstruction errors for static and movable parts. The latter two metrics are only applicable to methods with part segmentation capabilities, whereas A-SDF and CARTO are excluded due to their lack of part-aware modeling. Following Mu et al. (2021), for each surface, we sample 30,000 points and compute the bidirectional CD by averaging the distances from the prediction to the ground truth and from the ground truth to the prediction. The reported CD values are multiplied by 1,000.

**Pose and Joint Estimation Task.** We evaluate our framework against state-of-the-art pose estimators A-NCSH Li et al. (2020), GenPose Zhang et al. (2023), and ShapePose Zhou et al. (2025). We evaluate kinematic properties using four complementary metrics averaged over all parts: rotation error (Rot Err (°)), translation error (Trans Err (m)), joint state error (Err), and joint parameter errors (Ang Err (°), Pos Err (m)).

| Metrics | Methods | Reconstruction | | | | | Generation | | | | |
|---|---|---|---|---|---|---|---|---|---|---|---|
| | | Laptop | Eyeglasses | Dishwasher | Scissors | Drawer | Laptop | Eyeglasses | Dishwasher | Scissors | Drawer |
| CD-w↓ | A-SDF | 4.28 | 13.17 | 9.61 | 11.71 | 16.85 | 6.17 | 16.24 | 10.31 | 18.46 | 18.52 |
| | CARTO | 4.11 | 11.87 | 8.52 | 12.14 | 15.39 | 5.94 | 14.20 | 10.34 | 17.76 | 17.10 |
| | Paris | 2.53 | 7.84 | 13.42 | **5.83** | 14.93 | 3.76 | 9.33 | 15.51 | **7.45** | 18.15 |
| | Ditto | 2.81 | 9.36 | 12.50 | 10.69 | 17.12 | 3.33 | 11.14 | 13.22 | 12.14 | 19.73 |
| | KineDiff3D | **1.81** | **6.90** | **6.31** | 8.68 | **5.42** | **2.36** | **8.71** | **6.72** | 10.61 | **6.77** |
| CD-s↓ | Paris | 2.41 | 8.33 | 14.10 | **7.66** | 18.89 | 3.14 | 9.25 | 15.68 | **8.24** | 20.01 |
| | Ditto | 2.99 | 10.04 | 11.43 | 13.58 | 17.25 | 3.25 | 12.18 | 15.09 | 14.76 | 19.54 |
| | KineDiff3D | **2.21** | **7.82** | **7.41** | 10.21 | **6.13** | **2.13** | **8.63** | **8.24** | 12.13 | **8.41** |
| CD-m↓ | Paris | 3.32 | 8.27 | 15.22 | **8.05** | 21.44 | 4.86 | 12.32 | 16.17 | **9.20** | 26.14 |
| | Ditto | 3.66 | 13.40 | 14.75 | 12.07 | 19.36 | 4.25 | 16.19 | 17.22 | 12.68 | 32.15 |
| | KineDiff3D | **1.34** | **8.14** | **5.70** | 11.37 | **8.04** | **2.58** | **10.38** | **7.41** | 13.87 | **10.35** |

Table 1: **Comparison on the ArtImage Dataset.** Chamfer Distance (CD) results show KineDiff3D's superiority in both reconstruction and generation tasks. CD-w measures whole-object accuracy, CD-s static components, and CD-m movable parts. Note: A-SDF and CARTO lack segmentation capabilities - hence no CD-s/CD-m reported.

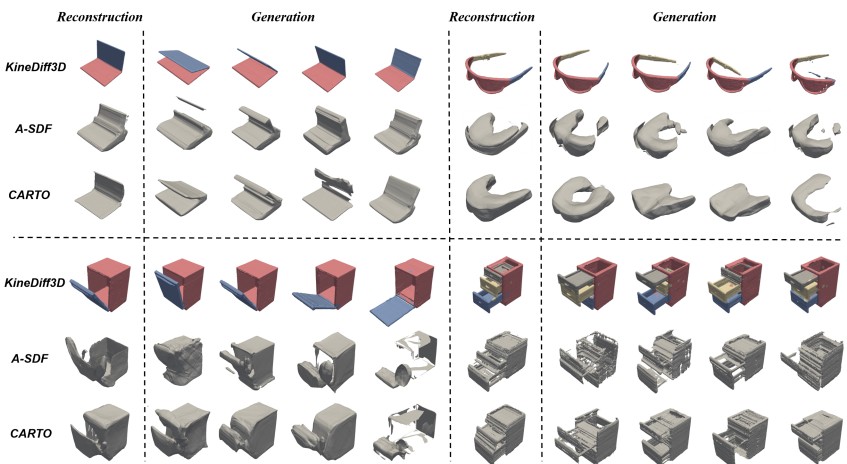

Figure 4: **Qualitative Results on the Dataset ArtImage.** Visual results demonstrate KineDiff3D's exceptional geometric fidelity in articulated object reconstruction and plausible generation, while baselines A-SDF and CARTO exhibit characteristic failures and lack segmentation capabilities.

## 4.3 COMPARISON WITH THE SOTA METHODS

**Mesh Reconstruction and Generation Performance.** The quantitative results, as presented in Table 1, demonstrate that our method consistently outperforms existing category-level reconstruction methods across all evaluated categories. For example, achieving significantly lower Chamfer Distance (CD-w=1.81 for laptop, -57% vs. A-SDF/CARTO). And our method enables precise part-aware reconstruction via its kinematic latent space, the qualitative results in Figure 4, quantifying separate static/movable part accuracy (CD-s=7.82, CD-m=8.14 for eyeglasses), outperforming Paris (8.33, 8.27) and Ditto(10.04, 13.40). Critically, KineDiff3D uniquely supports joint-conditioned generation of novel dynamic configurations. For Dishwasher, our generated configurations achieve CD-w=6.72, surpassing A-SDF (10.31) and CARTO (10.34), (CD-s=8.24, CD-m=7.41) surpassing Paris (15.68, 16.17) and Ditto(15.09, 17.22), while simultaneously preserving mechanical feasibility through kinematic constraints.

**Pose and Joint Estimation Performance.** We present the pose and joint estimation results of KineDiff3D on ArtImage in Table 2. Compared to classical methods, we achieve the best pose estimation results for the laptop category, with rotation error of $3.9°$. In Dishwasher, the translation error is only 0.052m. Concerning joint state error, we achieve a remarkable $4.9°$ for category eyeglasses. This superiority directly validates our method's effectiveness in integrating kinematic constraints during differentiable optimization. Qualitative results is provided in supplementary material.

## 4.4 ABLATION STUDY

**Self-occlusion Analysis.** To further examine the robustness of KineDiff3D under self-occlusion conditions, we categorized the test samples from the Dishwasher category into three subsets based on occlusion levels.

The occlusion level is quantified as the ratio of visible points to the total number of points, defining subsets with low (0%–40%), medium (40%–80%), and high (80%–100%) occlusion. As shown in Table 3 (I-III), the results show stable reconstruction and pose errors across increasing occlusion levels, robustly validating the efficacy of our method.

**Iterative Optimization Analysis.** We conducted an ablation study to evaluate the impact of our iterative optimization module. The results (Table 3 IV-VIII) demonstrate a clear monotonic improvement in both reconstruction accuracy (CD-w) and pose estimation (Rot Err, Trans Err) across optimization rounds.

The most significant gains occur within the first few iterations. For instance, CD-w improves from 7.82 (Round 1) to 6.56 (Round 3), while the rotation error is nearly halved. The optimization converges stably after 4 rounds, with subsequent steps yielding only marginal returns. This rapid convergence robustly validates the efficacy and efficiency of our joint-centric refinement strategy in bidirectionally minimizing errors and achieving kinematically consistent results. Please refer to supplementary materials for more ablation analysis.

| Category | Method | 6D Pose | | Joint State | Joint Parameter | |
|---|---|---|---|---|---|---|
| | | Rot Err (°) ↓ | Trans Err m ↓ | Err ↓ | Ang Err (°) ↓ | Pos Err (m) ↓ |
| Laptop | A-NCSH | 5.4 | 0.049 | 3.5° | 1.7 | 0.09 |
| | Genpose | 4.7 | 0.064 | 3.4° | 3.8 | 0.03 |
| | ShapePose | 4.8 | 0.058 | 5.9° | 3.3 | 0.06 |
| | **KineDiff3D** | **3.9** | **0.042** | **3.2°** | **0.9** | **0.03** |
| Eyeglasses | A-NCSH | 16.4 | 0.229 | 13.5° | 3.1 | 0.07 |
| | Genpose | 6.7 | 0.159 | 5.1° | 4.2 | 0.05 |
| | ShapePose | 5.4 | 0.088 | 5.7° | 3.9 | 0.07 |
| | **KineDiff3D** | **4.6** | **0.077** | **4.9°** | **1.8** | **0.03** |
| Dishwasher | A-NCSH | 4.4 | 0.091 | 3.8° | 6.1 | 0.11 |
| | Genpose | 6.2 | 0.140 | 3.8° | 4.8 | 0.09 |
| | ShapePose | 4.1 | 0.067 | 6.0° | 2.2 | 0.04 |
| | **KineDiff3D** | **3.3** | **0.052** | **2.3°** | **1.7** | **0.03** |
| Scissors | A-NCSH | **2.3** | 0.028 | 4.4° | 0.8 | 0.04 |
| | Genpose | 3.8 | 0.046 | 3.3° | 2.8 | 0.06 |
| | ShapePose | 2.6 | 0.039 | 4.2° | 1.9 | 0.08 |
| | **KineDiff3D** | 3.8 | **0.022** | **2.5°** | **0.5** | **0.02** |
| Drawer | A-NCSH | 3.3 | 0.108 | 0.41m | 3.5 | - |
| | Genpose | 4.4 | 0.128 | 0.12m | 3.3 | - |
| | ShapePose | 3.5 | 0.150 | 0.69m | 2.1 | - |
| | **KineDiff3D** | **2.8** | **0.072** | **0.57m** | **1.7** | - |

Table 2: **Comparison of pose and joint estimation** with State-of-the-arts on ArtImage Dataset.

| Index | Occlusion Level (Visibility) | Reconstruction(CD-w) | Rot Err (°) | Trans Err (m) |
|---|---|---|---|---|
| I | 0%-40% | 6.15 | 3.1 | 0.049 |
| II | 40%-80% | 6.34 | 3.4 | 0.051 |
| III | 80%-100% | 6.92 | 3.5 | 0.053 |
| **Index** | **Iteration Round** | **Reconstruction(CD-w)** | **Rot Err (°)** | **Trans Err (m)** |
| IV | 1 | 7.82 | 5.6 | 0.108 |
| V | 2 | 7.25 | 4.3 | 0.071 |
| VI | 3 | 6.56 | 3.7 | 0.058 |
| VII | 4 | 6.34 | 3.4 | 0.052 |
| VIII | 5 | 6.33 | 3.3 | 0.052 |

Table 3: **Ablation Study Results.** Note that experiments are on the category Dishwasher.

### 4.5 GENERALIZATION CAPACITY

**Experiments on Semi-Synthetic Scenarios.**
Table 4 (top) presents quantitative results on the ReArtMix dataset, showcasing KineDiff3D's robust performance in semi-synthetic scenarios. Our method achieves a CD-w of 0.92 for reconstruction tasks in Box. Furthermore, for generation tasks, KineDiff3D records a CD-s and CD-w of 3.01 and 2.72 in Scissors. Qualitative results can be seen in Figure 5 (top).

**Test on Real-world Scenarios.** We evaluate KineDiff3D's generalization using real-world depth images from the RBO dataset Martín-Martín et al. (2018), following A-SDF's protocol Mu et al. (2021). Table 4 (bottom) shows our synthesis-trained model achieves (CD-w: 2.56, CD-s: 2.74 and CD-m: 2.71 in laptop) for reconstruction, demonstrating robust cross-domain transfer without real-world training. Using KA-VAE's latent space, we generate dynamic shapes by adjusting joint angles while preserving geometry codes, yielding plausible configurations for unseen articulation states (CD-w: 3.46, CD-s: 3.85, CD-m: 3.72). Qualitative results can be seen in Figure 5 (bottom).

| Category | Reconstruction | | | Generation | | | 6D Pose | |
|---|---|---|---|---|---|---|---|---|
| | CD-w↓ | CD-s↓ | CD-m↓ | CD-w↓ | CD-s↓ | CD-m↓ | Rot Err (°) ↓ | Trans Err (m) ↓ |
| | | | | ReArtMix (Semi-Synthetic Scenarios.) | | | | |
| Box | 0.92 | 1.31 | 1.46 | 1.85 | 2.02 | 2.41 | 2.9 | 0.016 |
| Stapler | 2.00 | 2.54 | 2.31 | 2.71 | 3.42 | 3.67 | 2.9 | 0.029 |
| Cutter | 2.65 | 2.71 | 3.11 | 3.45 | 3.66 | 3.78 | 2.4 | 0.016 |
| Scissors | 1.81 | 2.53 | 2.45 | 2.37 | 3.01 | 2.72 | 3.9 | 0.018 |
| Drawer | 1.32 | 1.75 | 2.04 | 1.73 | 1.86 | 2.94 | 1.6 | 0.017 |
| | | | | RBO (Real-world Scenarios.) | | | | |
| Laptop | 2.56 | 2.74 | 2.71 | 3.46 | 3.85 | 3.72 | 4.6 | 0.062 |
| Box | 1.73 | 1.97 | 2.34 | 2.77 | 3.15 | 3.61 | 4.4 | 0.047 |

Table 4: **Quantitative Results on the Semi-Synthetic and Real-world Datasets.**

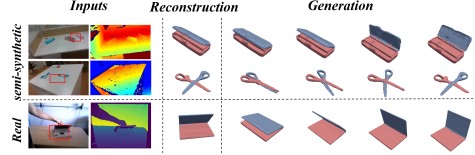

Figure 5: **Qualitative Results on the Semi-Synthetic and Real-world Datasets.**

## 5 CONCLUSION

In this paper, we propose KineDiff3D, a novel framework for category-level articulated object shape reconstruction and generation. By integrating a Kinematic-Aware Variational Autoencoder (KA-VAE), conditional diffusion modeling, and an iterative optimization strategy, our approach achieves superior performance in both geometric fidelity and kinematic accuracy compared to existing methods. Extensive experiments on synthetic, semi-synthetic, and real-world datasets demonstrate KineDiff3D's ability to generalize across diverse object categories and handle severe occlusions.

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
