# OpenReview forum: "KineDiff3D: Kinematic-Aware Diffusion for Category-Level Articulated Object Shape Reconstruction and Generation"
_ICLR.cc/2026/Conference — ICLR 2026 Conference Withdrawn Submission_

### Official Review · Reviewer_wFXX · 2025-10-28

**Soundness:** 2
**Presentation:** 2
**Contribution:** 2
**Rating:** 2
**Confidence:** 3

**Summary:**

The paper propose a Kinematic-Aware diffusion for category-level articulated object shape reconstruction and generation. It first encode SDFs, joint angles, and part segmentation into a structured latent space via a Kinematic-Aware VAE and then employ two conditional diffusion models for regressing global pose and joint parameters. Finally, it produce an iterative optimization module to refine reconstruction.

**Strengths:**

1. The idea of encoding everything including the geometry and kinematic informations into a unified latent space sounds reasonable to me, since the development of 3D generation models gradually switch to native 3D space.
2. The two diffusion models that respectively learns kinematic-aware informations and part geometry sounds reasonable.

**Weaknesses:**

1. The way the authors cite papers is really hard for reading, which I believe is due to the package or template.
2. The authors should polish the figures, especially Fig. 2. In the Pose and Joint Estimation Module, what's the difference between the two lines with (X_T, Y_T)? Does that mean a single inference step? If I understand it correctly, this is the part of a conditional diffusion model that conditions on the partial point cloud (encoded by PointNet++) and predicts base pose and joint parameters but the way the authors draw the figure is quite confusing. What is actually been denoising?
3. Does the Pose and Joint Estimation Module and the reconstruction module related with each other? From the reviewer's understanding, it seems that these two modules are separate ones.
4. The authors should elaborate on the generation mentioned in all the experiments. It seems that they are mainly generating novel poses of the objects instead of generating articulated objects from few images?
5. The original PARIS method assumes images from two-state but from the reviewer's understanding, it seems that the proposed method only takes in single-state images as input? How is the comparison performed?

I appreciate to directly denoise in a native VAE space, but the writing and explanation seems confusing, so I currently leans towards rejecting. But I am willing to change my view after the rebuttal and see other reviewers' comments.

**Questions:**

On what dataset are the KA-VAE and the diffusion models trained?

---

### Official Review · Reviewer_pmJX · 2025-10-28

**Soundness:** 2
**Presentation:** 2
**Contribution:** 2
**Rating:** 4
**Confidence:** 4

**Summary:**

This paper presents KineDiff3D, a diffusion-based framework for reconstructing articulated 3D objects from sparse multi-view or single-view inputs. The approach achieves state-of-the-art results on benchmarks like PartNet-Mobility and new synthetic data.

**Strengths:**

- Novel integration of kinematic constraints into diffusion-based 3D modeling.

- Demonstrates improved generalization to unseen articulations and novel part combinations.

- Visualization and ablations clearly illustrate the role of kinematic priors.

**Weaknesses:**

- The writing of the paper needs improvement, and the overview figure cannot present the methods clearly.

- Some improvement margins over baselines are modest, suggesting incremental benefit in certain settings. Moreover, it seems that it is not compared with the latest SOTA methods, but only with those from a few years ago.

- Limited qualitative demonstrations on real-world data; most results are synthetic.

- The novelty mainly lies in integrating existing techniques (diffusion + kinematic loss) rather than introducing a fundamentally new formulation.

- It seems rather strange that the ablation experiment is only conducted in one category (Dishwasher).  How can experiments in one category prove that these components are also useful in other categories?

**Questions:**

see weakness

---

### Official Review · Reviewer_yMLM · 2025-10-30

**Soundness:** 2
**Presentation:** 1
**Contribution:** 2
**Rating:** 4
**Confidence:** 4

**Summary:**

This work tackles 3D reconstruction and pose estimation of articulated objects from single-view inputs. The proposed KineDiff3D framework uses a Kinematic-Aware VAE to encode geometry, joint angles, and segmentation, along with conditional diffusion models for pose/joint regression and latent code generation. An iterative optimization module further refines results while maintaining articulation constraints. Experiments demonstrate strong performance across multiple datasets.

**Strengths:**

1. Comprehensive unified framework: The paper presents a well-designed end-to-end system that jointly addresses multiple challenging tasks—shape reconstruction, pose estimation, and novel articulation generation—within a single framework.

2. The bidirectional optimization module that simultaneously refines reconstruction accuracy and kinematic parameters while preserving articulation constraints works well. This design leverages the mutual dependencies between geometry and kinematics, likely leading to more robust and accurate results compared to methods that treat these aspects independently.

**Weaknesses:**

1. The inputs to the model should be clarified at the beginning of the method section. Specifically:

a) Is the input a single-view image with depth information?

b) How is the full object point cloud (shown at the top of Figure 2) obtained?

c) How is the partial object point cloud obtained?

2. Why did you choose to use PointNet and PointNet++ as there are many more powerful models?

3. The pipeline overview in Figure 2 needs improvement. The flow lines are difficult to follow and make the overall process unclear.

4. The glasses' legs appear incomplete in the generated novel articulations in Figure 4. Could you clarify why this occurs? I think that novel articulation generation should preserve the object's shape and only modify its pose configuration.

5. Appendix formatting (Section 4.1): When using `Metrics` as a paragraph. I think that  "Reconstruction and Generation Task" and "Pose and Joint Estimation Task" should not be listed as paragraphs.

**Questions:**

Please see the weaknesses. I think the writing is not friendly for readers, especially the method section. It would be better if the authors could further improve it.

---

### Official Review · Reviewer_dy5t · 2025-11-01

**Soundness:** 2
**Presentation:** 2
**Contribution:** 2
**Rating:** 2
**Confidence:** 3

**Summary:**

The paper presents KineDiff3D, a single-view articulated object reconstruction framework that integrates a Kinematic-Aware VAE (KA-VAE), conditional diffusion models, and an iterative optimization module.
The pipeline first encodes geometry (SDF), part segmentation, and joint angles into a shared latent space using KA-VAE. Two diffusion models are then trained: one for pose/joint estimation, and another for latent code generation from partial point clouds. Finally, a joint-centric optimization loop refines pose and geometry via Chamfer distance while preserving kinematic constraints.
Experiments on synthetic, semi-synthetic, and real datasets (ArtImage, ReArtMix, RBO) show that KineDiff3D improves Chamfer Distance and joint pose accuracy over previous category-level baselines such as A-SDF, CARTO, Paris, and Ditto.

**Strengths:**

The paper tackles an important and challenging problem — category-level articulated object reconstruction from single views.

Integration of geometry, kinematics, and generative modeling within a single framework is conceptually appealing.

The ablation on iterative optimization shows the model can improve with refinement steps.

The implementation appears complete and reproducible in principle, showing non-trivial engineering effort.

**Weaknesses:**

Lack of true novelty: The proposed KA-VAE and diffusion combination is a straightforward hybrid of known components. Recent works (Real2Code 2024, Reacto 2024, ArticulatedGS 2025) already address similar goals with more rigorous modeling and stronger baselines.

Misleading claim of “generation”: The paper never performs unconditional or cross-category generation; it only interpolates joint angles of known shapes.

Incomplete evaluation: The experiments omit essential baselines, use limited datasets, and report no standard deviations or statistical tests.

Unclear articulation encoding: Handling of variable joint topology, parameterization of revolute/prismatic joints, and fusion between geometry and motion are insufficiently explained.

Superficial diffusion analysis: There is no comparison showing that diffusion improves over direct latent regression or normalizing flows.

Overly dense presentation: The paper reuses long derivations of SDE-based diffusion without insight, making it difficult to distinguish novelty from background.

No qualitative failure analysis or runtime discussion, despite introducing multiple heavy submodules (two diffusion networks + VAE + optimization).

**Questions:**

What is the exact advantage of diffusion over a simple regression network for latent prediction? Can you provide quantitative evidence (e.g., ablation replacing diffusion with MLP)?

How does the framework handle variable joint topology across categories (e.g., eyeglasses vs. drawer)? Are joint parameters padded, masked, or predicted per-category?

Is the KA-VAE trained jointly with the diffusion modules or sequentially? If sequential, how is latent-space alignment ensured?

How many iterations are required for the optimization loop, and what is its runtime overhead?

How are “generation” results produced? Are they stochastic samples from diffusion or deterministic angle interpolation?

Could the authors compare against 2024–2025 state-of-the-art methods (e.g., Reacto, Real2Code, ArticulatedGS) using the same metrics and datasets to substantiate the claimed progress?

Please clarify whether the real-world evaluation uses any domain adaptation or fine-tuning, as results seem unexpectedly strong given the fully synthetic training.

---

### Note · Authors · 2025-12-01

I have read and agree with the venue's withdrawal policy on behalf of myself and my co-authors.